biomechanics/mechanical engineering/power and energy systems

gait, load carriage, energetics, biomechanical energy harvesting

**Author for correspondence:**
Jean-Paul Martin
e-mail: 8jpm4@queensu.ca

# Generating electricity while walking with a medial–lateral oscillating load carriage device

Jean-Paul Martin[1,2] and Qingguo Li[1,2]

[1]Bio-Mechatronics and Robotics Laboratory, Mechanical and Materials Engineering, and [2]Ingenuity Labs, Queen's University, Kingston, Canada K7L 3N6

J-PM, 0000-0002-9105-0008

Biomechanical energy harvesters generate electricity, from human movement, to power portable electronics. We developed an energy harvesting module to be used in conjunction with a load carriage device that allows carried mass in a backpack to oscillate in the medial–lateral (M–L) direction. The energy harvesting module was designed to tune M–L oscillations of the carried mass to create favourable device–user interaction. We tested seven energy harvesting conditions and compared them to walking with the device when the weight was rigidly fixed to the backpack frame. For each energy harvesting condition, we changed the external load resistance: altering how much electricity was being generated and how much the carried mass would oscillate. We then correlated device behaviour to the biomechanical response of the user. The energy harvesting load carriage system generated electricity with no significant increase in the metabolic power required to walk, when compared to walking with the carried weight rigidly fixed. The device was able to generate up to $0.22 \pm 0.03$ W of electricity, while walking with 9 kg of carried weight. The device also reduced the interaction forces experienced by the user, in the M–L direction, compared to walking with the device when the mass was rigidly fixed.

## 1. Introduction

The ubiquity of portable electronics, and our reliance on them, are limited by maintaining a continual power source for their operation. This is particularly an issue for those who work in remote areas with no access to the power grid. Military, field scientists, disaster relief workers and recreational enthusiasts all rely on electronics such as global positioning devices, satellite phones, avalanche beacons or scientific equipment. These portable electronics are almost exclusively powered with batteries. However, total energy in a battery is a function of its

weight and size [1]. When considering trips of extended duration in remote areas, the weight and size of batteries required for the entire trip duration can be impractical to carry.

Biomechanical energy harvesters (BMEHs) have been identified as being a viable solution to providing a renewable energy source for portable electronics [2–6]. BMEHs generate electricity from human movement, as the wearer goes about their day. The advantage of a BMEHs is that they do not require dedicated attention for their operation, like a hand crank generator. A particular subset of BMEHs, called inertial based harvesters, generate electricity from the movement of an external mass [7–17]. This subset of harvesters are particularly advantaged by the aforementioned groups who work in remote areas, as these individuals often carry weight in a backpack for their expedition, ideal for use as a proof mass in an energy harvesting system. Rome et al. [7] were the first to develop an energy harvesting backpack that generated electricity from the vertical oscillations of carried weight in a backpack. Their device generated up to 7.4 W while carrying 38 kg of weight [7]. Since then, various groups have created energy harvesting backpacks that similarly generate electricity (46 mW−10.6 W) [9,15] from allowing the carried mass to oscillate in the vertical direction [9,13–16]. Of the aforementioned devices, none have targeted medial−lateral (M−L) oscillations of carried mass in a backpack. The development of such could improve load carriage for users by creating favourable device−user interaction, while simultaneously generating energy for portable electronics.

While walking, the trunk oscillates in both M−L and vertical directions. Recently, we developed a device that allows the carried weight in a backpack to oscillate in the M−L direction and studied its effects on gait [18]. Our device showed that it was able to reduce the interaction force between device and user when the carried weight was oscillating, compared to fixed. However, the metabolic power required to walk with the device when oscillating was higher than that for the fixed condition. Previous work has shown that the metabolic cost of walking can be reduced by providing an M−L restorative force acting on the subject's pelvis [19–21]. As well, modelling work has also shown that walking economy can be improved by having the walking surface oscillate and do work on the walking model [22]. Since the carried mass in our M−L load carriage device was oscillating out-of-phase, the M−L interaction force component was also out-of-phase, resembling a restorative force to the centreline similar to studies with elastically tethered subjects [19–21]. Therefore, a possible reason for why there was an increase in the metabolic energy required to walk with the device when oscillating was the large M−L excursions of the carried mass. With greater lateral oscillations, we observed a greater frontal plane moment developed about the backpack's origin, transferred to the subject via backpack attachment points. Decreased mass oscillation amplitude was identified as being a solution for decreasing the frontal plane moment in our previous experiment [18]. To remedy the oscillation amplitude, an energy harvesting module was developed to alter damping of the system, to reduce mass oscillations, and simultaneously generate electricity. The energy harvesting module will be used to tune M−L oscillations, to create favourable device−user interactions.

The following paper presents the design and testing of a biomechanical energy harvesting backpack that allows the carried mass to oscillate in the M−L direction. The purpose of this study is to understand the effects of electromagnetic damping on M−L oscillating load carriage, device−user interactions, walking performance and electrical power generation. To investigate this, subjects walked with the device under varying amounts of damping present, as dictated by the energy harvesting module. The condition with the least amount of damping present is the open circuit condition (open), where no energy is being harvested. Then, six energy harvesting conditions are tested (R1−R6). R1 has the least amount of damping present and R6 has the greatest amount of damping present. All harvesting conditions were compared to a fixed condition (fixed), where the oscillating mass was rigidly connected to the backpack frame. We hypothesize, from preliminary modelling results of the backpack device, that lower damping ratios (R1, R2) will produce the most electricity and will decrease with increased damping (R3 to R6). We hypothesize that with increased damping from energy harvesting, we will observe a reduction in the frontal plane interaction moment, which will coincide with a reduction in the metabolic cost of walking. This is because with increased damping, modelling predicts a decrease in the amplitude of oscillations of the carried mass.

## 2. Methods

### 2.1. Energy harvesting backpack

The energy harvesting backpack consists of an M−L load carriage device, described in [18], and an energy harvesting module (figure 1). The medial-lateral load carriage device uses cylindrical weights,

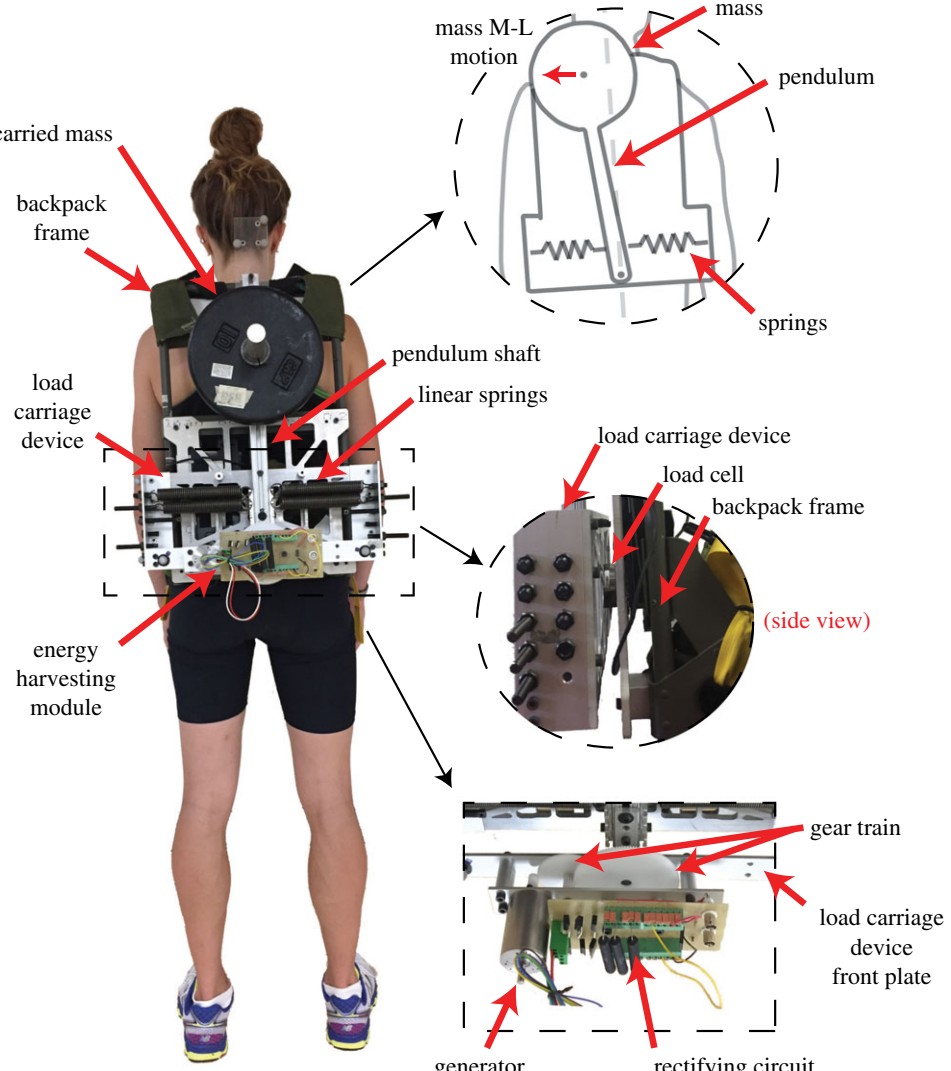

**Figure 1.** A posterior view of the energy harvesting backpack worn by a subject. (top-right blowout) A simplified diagram of the device being worn by a user. The structure of the device is simplified to demonstrate critical components. Carried mass motion in the M−L direction is indicated as a red arrow. (middle-right blowout) A side view showing where the load carriage device mounts to the backpack frame via the load cell. (bottom-right blowout) A view from above of the energy harvesting module attached to the front of the M−L load carriage device.

simulating carried weight (mass = 9 kg), suspended on an inverted pendulum (length = 30 cm). Linear springs, acting up the length of the inverted pendulum shaft in increments of 2 cm, provide a restorative force on the inverted pendulum. Both pendulum and springs are housed within a rectangular frame (weight = 2.2 kg) (table 1). The rectangular frame is mounted to a commercially available backpack frame (weight 2.4 kg) via a 6 d.f. load cell (Mini 45, ATI Industrial Automation, Apex, NC) (weight = 92 g). The energy harvesting module (weight = 0.6 kg) attaches to the front plate of the device and consists of a two-stage gear train, an electromagnetic generator (EC-i 100W, Maxon, Switzerland), and a rectifying circuit (figure 1). The first stage of the gear train (ratio = 10) is attached to the main drive shaft of the inverted pendulum. The second stage (ratio = 6.7) attaches to the drive shaft of the generator. An overall gear ratio of 67:1 was chosen to maximize the gear ratio, given the largest commercially available polyacetal gears that also met our space and strength requirements (Misumi USA, IL). A greater gear ratio was determined from modelling to minimize losses due to rotor winding resistance. The three-phase alternating current of the generator is full wave rectified using Schottky diodes (STPS20M60, STMicroelectronics, Switzerland). Electrical resistors are used in different configurations to vary the amount of electricity harvested and the amount damping present. By reducing the value of the electrical resistors in the circuit, the generator harvests more electricity

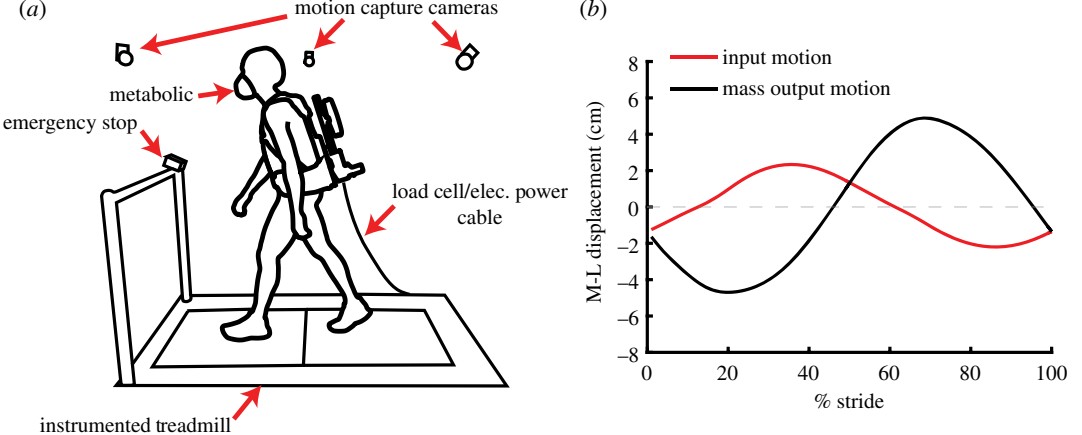

**Figure 2.** (*a*) The experimental set-up used in testing. (*b*) Input M−L displacement of the backpack frame and output M−L relative displacement of the carried mass, as a function of % gait cycle (open circuit condition). Mass M−L displacement is relative to the backpack frame's centreline. Waveforms demonstrate how mass oscillations are near out-of-phase with input motions of the backpack frame with a chosen frequency ratio of 0.67. Each waveform represents the average across all subjects.

**Table 1.** Total device mass and masses of individual parts of the load carriage device.

| component | weight (kg) |
| --- | --- |
| load carriage device | 2.2 |
| backpack frame | 2.4 |
| load cell | 0.6 |
| harvesting module | 0.6 |
| total | 5.8 |

from the angular velocity of the pendulum. However, this also applies more back electromotive torque on the pendulum, which increases the overall damping of the system and leads to decreased mass oscillations. Therefore, maximum electrical power harvested during walking is a balance between harvesting as much electricity from the oscillations of the pendulum without damping the pendulum's oscillations to a suboptimal point.

## 2.2. Experimental protocol

Eleven ($n = 11$) healthy, young adults, with no known prior gait pathologies participated in the study (height $= 181.7 \pm 9.6$ cm, weight $= 73.2 \pm 9.8$ kg, age $= 23.7 \pm 2.3$ years, males $= 8$, females $= 3$). All subjects gave informed consent. The experiment was approved by the General Research Ethics Board of Queen's University (Ref. No. 28210).

The experimental set-up is shown in figure 2*a*. Subjects first conducted an acclimatization session 24−48 h prior to the experiment. Subjects walked with the device for 17 min in total to familiarize themselves with the device and treadmill (AMTI Force-Sensing Tandem Treadmill, AMTI Inc., MA). Treadmill walking speed was held at a constant $1.3 \text{ m s}^{-1}$ for all experiments. The 17 min of acclimatization consisted of the subject walking 2 min with the carried mass rigidly fixed, 5 min in an oscillating condition with an external load resistance that resulted in an equivalent modelled damping ratio of 0.7, 5 min with an equivalent modelled damping ratio of 0.04, and 5 min with an open circuit (no electrical resistance). Heel strike events were recorded for the last 1 min of the 3 oscillating conditions to determine the subject's preferred stride frequency. The subject's stride frequency was used to determine the equivalent torsional spring constant for experiments. A spring constant was chosen such that the modelled undamped natural frequency of the inverted pendulum was 0.67 times the stride frequency of the subject. A natural frequency less than the stride frequency is required for out-of-phase oscillations of the carried mass with respect to the input motion of the trunk (figure 2*b*). A ratio of 0.67 was chosen to ensure out-of-phase oscillations in the presence of additional compliance added from the backpack to human interface.

On experimental day, the subject's resting metabolic rate while standing was measured for 6 min. Then, subjects walked 6 min with the device to warm up. For the first 3 min of warm up, subjects walked with the highest damped condition used in testing (R6). Then, for the second 3 min, subjects walked with the lowest damped condition used in testing (open condition—no electrical resistance). Subjects performed eight conditions, in random order, of 6 min in length. The conditions are as follows: fixed, open circuit (open) and equivalent damping ratios of $\zeta = 0.04$ (R1), $\zeta = 0.16$ (R2), $\zeta = 0.24$ (R3), $\zeta = 0.36$ (R4), $\zeta = 0.5$ (R5), $\zeta = 0.7$ (R6). The damping ratio, $\zeta$, is defined as the ratio of the modelled damping of the system to the modelled critical damping for the system. The damping ratios were chosen from a combination of modelling, that determined the damping ratio estimated to generate the maximum electrical power, and pilot testing. Based on the generator model, we selected configurations of external load resistances that were predicted to result in the prescribed damping ratio. Kinematics, kinetics, device mechanics and metabolic data were collected over the last 2 min of each 6 min trial.

## 2.3. Data measurement and analysis

### 2.3.1. Device mechanics

Pendulum angle was estimated using motion capture (Oqus, Qualysis, Sweden). Motion capture was sampled at 200 Hz and filtered using a second-order low-pass zero-phase Butterworth filter with cut-off frequency of 8 Hz. A local coordinate frame was defined on both backpack frame and pendulum shaft using reflective markers. Mass relative M–L displacement was defined as the displacement of the mass with respect to a local vertical axis located at the pendulum origin. Mass relative M–L displacement was measured over the last 2 min of each trial. A discrete Fourier transform was performed to determine the amplitude and phase of oscillations. Mass oscillation frequency was determined to be the frequency bin with greatest amplitude, which was verified by comparing with the stepping frequency from heel strike detection. The amplitude ratio, or the ratio of M–L linear displacement amplitude of the carried mass to the M–L linear displacement amplitude of backpack frame input motion, was determined as a non-dimensional measure of mass oscillations. The phase angle of mass oscillations were determined relative to input M–L displacement of the backpack frame. This was estimated by subtracting the phase angle of input motion from the phase angle of pendulum oscillations.

Device interaction force and moments were measured over the last 2 min of each trial using a 6 d.f. load cell. Force and moment were sampled at 1000 Hz and filtered using a low-pass zero-phase Butterworth filter with cut-off frequency of 16 Hz. The peak M–L and vertical interaction force, and frontal plane interaction moment, were defined as the maximum magnitude of force/moment over each stride, averaged over the last 2 min of each condition. The interaction force measured by the load cell was assumed to be the interaction force experienced by the user. We did not account for the compliance and load transfer from the backpack frame, through the straps, to the user.

The electrical power harvested was determined by measuring line current and voltage drop over the external load. The line current was measured using a current sensing resistor. The voltage drop over the external load was determined as the sum of voltage drop over the current sensing resistor and configurable external load resistance. Both were sampled at 1000 Hz and filtered using a low-pass zero-phase Butterworth filter with cut-off frequency of 16 Hz. The product was determined to be the instantaneous electrical power. Average electrical power was determined by integrating the instantaneous electrical power over the last 2 min of each trial, which was then divided by the sampling period.

### 2.3.2. Metabolic cost

The metabolic power of walking was estimated over the last 2 min of each trial using an open respirometry unit (K4b2, Cosmed, Italy). The average gross metabolic power was calculated using the standard equation from Brockway [23]. The net metabolic power was calculated by subtracting the resting metabolic rate, which was then divided by the subject's body mass.

### 2.3.3. Lower-limb mechanics

Kinematics and kinetics of the right lower limb were determined over 10 consecutive strides over the last 2 min of each trial. Joint angles, moments, power and work were calculated using a custom Matlab script

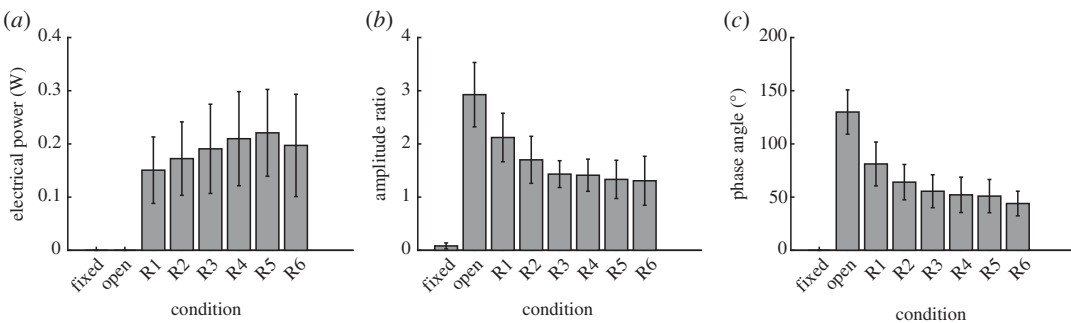

**Figure 3.** Device response. (*a*) Electrical power harvested for each condition. (*b*) The amplitude ratio of carried mass oscillations, across all backpack conditions. (*c*) The phase angle of carried mass oscillations, with respect to M−L input excursions, across all conditions.

using methods described in [24]. Motion capture was sampled at 200 Hz and was filtered using a second-order low-pass zero-phase Butterworth filter with cut-off frequency of 8 Hz. Three-dimensional ground reaction forces were measured at 1000 Hz and filtered using a second-order low-pass zero-phase Butterworth filter with cut-off frequency at 32 Hz. Motion capture marker locations and limb segment definitions are from a modified method as described in [24]. The modification was required because the backpack's waist straps covered anatomical locations of the anterior and posterior superior iliac spine. Therefore, we assumed that the trunk and pelvis were a single rigid body segment and tracked the pelvis using markers on the trunk (C7, sternal notch, right and left acromion markers). A transformation matrix relating the pelvis pose, with respect to the trunk's marker set, was determined with a static pose without the backpack donned. Segment mass centre and the moment of inertia of limb segments were determined using [25].

### 2.3.4. Spatial-temporal gait parameters

Stride frequency was defined as the period between consecutive right heel strikes, averaged over the last 2 min of each trial. Step length and width were defined as the M−L and for-aft distances between calcaneus markers at heel strike, averaged over the last 2 min. Step width and length variability are defined as the standard deviation of step width and length, respectively, over the last 2 min of each condition.

### 2.3.5. Statistical tests

Statistical tests were performed in Matlab for the amplitude ratio, phase angle, device interaction force and moment, electrical power, metabolic expenditure, joint range of motion, joint peak power, joint work, stride frequency, step length and step width. A one-way repeated measures analysis of variance model was used, where backpack condition (fixed, open, R1, R2, R3, R4, R5, R6) was the within-subject variable ($\alpha < 0.05$). Sphericity was tested using Mauchly's, and when violated, a Greenhouse–Geisser or Huynh–Feldt correction was made. *Post hoc* analysis was performed using Tukey's test. Joint power, work, stride frequency, step length and step width were omitted for all conditions of a single subject due to issues with the instrumented treadmill.

# 3. Results

## 3.1. Device mechanics

The energy harvesting backpack generated an average $0.19 \pm 0.03$ W during walking (figure 3*a*). The condition with the greatest amount of electricity harvested was the R5 condition ($0.22 \pm 0.03$ W). Peak electrical power was shifted to a damping ratio greater than expected (modelled peak = R2), before decreasing with increased damping. This suggests that the balance between harvesting the most electricity from pendulum oscillations, without damping the pendulum's oscillations to a suboptimal point, occurs at higher damping ratios than what was predicted by the model. A large intra-subject

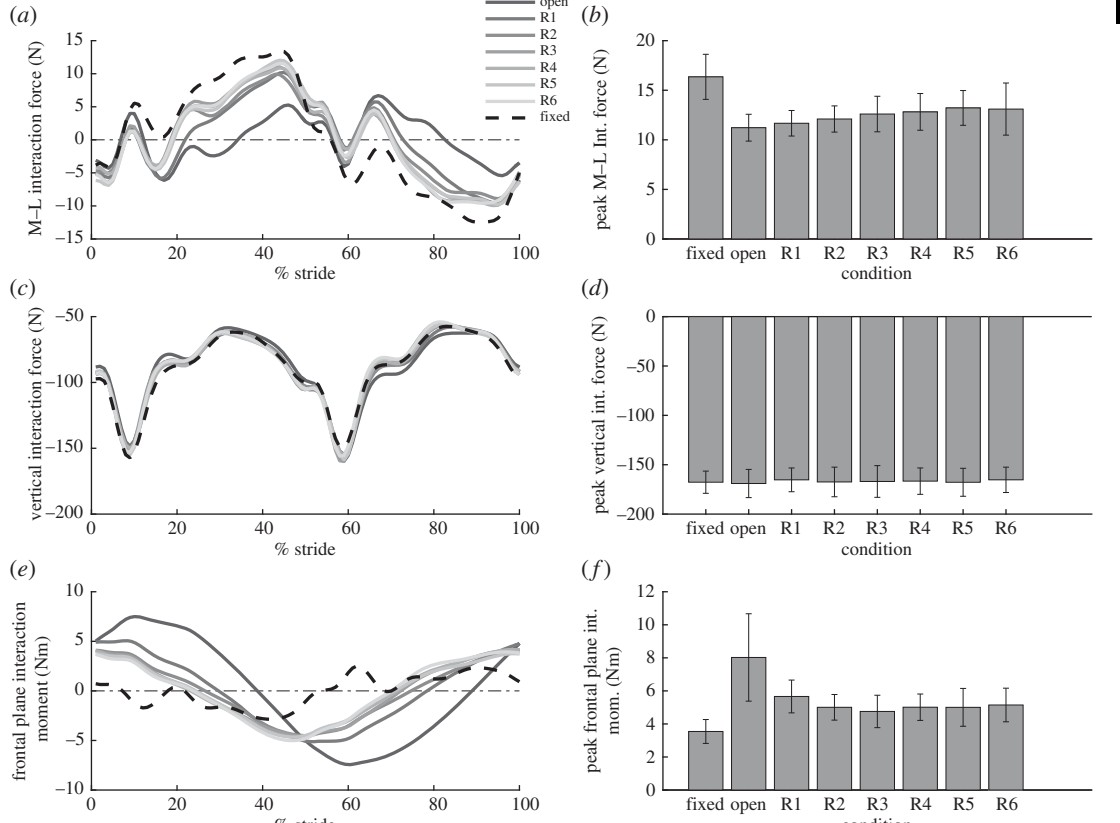

**Figure 4.** Interaction forces and moments between device and user. (a) The M−L interaction force waveform, measured by the load cell, as a function of % gait cycle. Increased transparency indicates an increase in damping from the electromagnetic generator. Each waveform represents the average across all subjects for that condition. (b) The peak M−L interaction force, averaged across subjects, for each backpack condition. (c) The vertical interaction force waveform as a function of % gait cycle. (d) The peak magnitude of vertical interaction force across all conditions. (e) The frontal plane interaction moment waveform as a function of % gait cycle. (f) The peak magnitude of the frontal plane interaction moment.

variability was observed for electrical power harvested, where greater trunk M−L excursions tended to result in greater electricity harvested.

The amplitude ratio, as a function of backpack condition, is shown in figure 3b. As expected, increased damping from energy harvesting decreased mass oscillation amplitude. This indicates that the energy harvesting module can, in fact, be used as an effective control of device behaviour. However, the amplitude ratio did not significantly decrease from the R3 to R6 condition (pairwise comparison of R3–R4: $p = 1$, R3–R5: $p = 0.9$, R3–R6: $p = 0.8$). Small motions were measured during the fixed condition. This was assumed to be due to the compliance of the device frame, pendulum and bolting system used to fix the pendulum.

The phase angle of the carried mass, with respect to M−L input motion, is shown as a function of backpack condition in figure 3c. As expected, the phase of mass oscillations approached in-phase behaviour as damping increased. During the open condition, with damping from mechanical losses only, the phase angle of mass oscillations were most out-of-phase with respect to M−L input motion ($130.0 \pm 20.8°$).

The peak M−L interaction force was significantly reduced during oscillating conditions, compared to fixed, by $32 \pm 9\%$, $32 \pm 6\%$, $28 \pm 5\%$, $25 \pm 6\%$, $24 \pm 4\%$, $21 \pm 4\%$ and $24 \pm 8\%$ for the open, R1, R2, R3, R4, R5 and R6 conditions, respectively ($p = 0.04, 0.04, 0.03, 0.02, 0.003, 0.006, 0.01$) (figure 4b). As expected, the open condition resulted in the greatest decrease in peak M−L interaction force, compared to fixed, with reductions decreasing as the damping from energy harvesting increased. However, even in the presence of energy harvesting, the load carriage device reduced the peak M−L interaction force experienced by the user during load carriage. There was no significant difference in peak vertical force between any backpack condition ($p = 0.7$) (figure 4d).

The peak frontal plane interaction moment was found to significantly increase from the fixed condition to the open condition by $53 \pm 10\%$ ($p = 0.01$) (figure 4f). This was due to increased mass

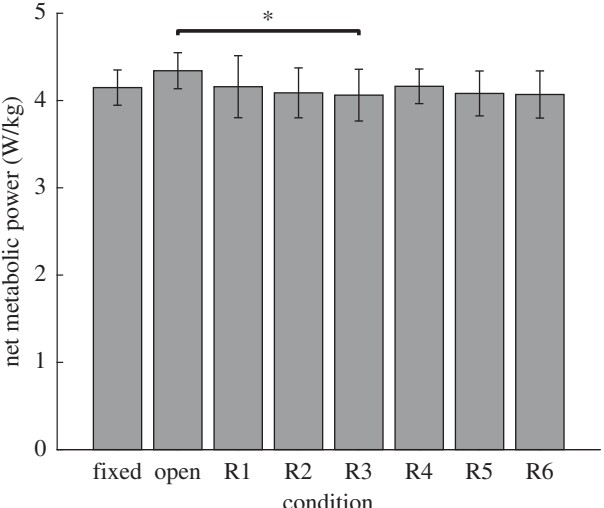

**Figure 5.** The net metabolic power of walking during each backpack condition. An asterisk overhead indicates a significant difference between conditions.

oscillation amplitude in the absence of damping from energy harvesting. As expected, in the presence of damping, the peak frontal plane interaction moment decreased. There was a significant decrease in the peak moment during the R3 condition compared to the open ($p = 0.04$) and R1 ($p = 0.04$) condition. As well, there was a significant decrease in the peak moment in the R2 condition compared to the open condition ($p = 0.04$). However, there was still a significant increase from fixed by $36 \pm 8\%$, $30 \pm 9\%$, $24 \pm 16\%$, $29 \pm 6\%$, $26 \pm 15\%$, $31 \pm 6\%$ for the R1, R2, R3, R4, R5 and R6 conditions, respectively, ($p = 0.001, 0.006, 0.01, 0.0004, 0.003, 0.0005$).

### 3.2. Metabolic power

The metabolic power while walking with the device in all conditions is shown in figure 5. The metabolic power ($4.3 \pm 0.2 \text{ W kg}^{-1}$) was greatest during the open condition. The metabolic power ($4.1 \pm 0.3 \text{ W kg}^{-1}$) was the lowest for the R3 condition, and was significantly lower than the metabolic power in the open condition ($p = 0.004$). These results indicate that with regulated damping, we can achieve better device–user interaction, resulting in a decrease in metabolic power while walking with the device. There was no significant difference between the metabolic power for the fixed condition, and any condition with energy harvesting (Fixed-R1: $p = 1$, Fixed-R2: $p = 0.9$, Fixed-R3: $p = 0.7$, Fixed-R4: $p = 1$, Fixed-R5: $p = 0.9$, Fixed-R6: $p = 0.6$).

### 3.3. Lower-limb mechanics

Lower-limb joint angles in the sagittal and frontal plane are shown in figure 6. We observed a significant increase in the hip range-of-motion, in the frontal plane (abd/add), from the fixed condition to the open condition ($p = 0.04$) (figure 6f). However, the increase in range-of-motion from fixed tended to decrease when damping from energy harvesting was present (Fixed-R1: $p = 0.3$, Fixed-R2: $p = 0.3$, Fixed-R3: $p = 0.7$, Fixed-R4: $p = 0.1$, Fixed-R5: $p = 0.8$, Fixed-R6: $p = 0.1$).

Lower-limb joint power is shown for all backpack conditions in figure 7. Although there was a slight decrease in the second peak positive hip power for the open condition, this difference was found to be not significant compared to the fixed condition ($p = 0.8$) (figure 7c). There were no significant differences between backpack conditions for peak power of the ankle (positive peak at push off: $p = 0.5$), knee (negative peak: $p = 0.7$) or hip (early stance positive peak: $p = 0.7$, stance negative peak: $p = 0.8$). As well, there was no significant difference observed in work done by the ankle (positive: $p = 0.8$, negative: $p = 0.6$), knee (positive: $p = 0.6$, negative: $p = 0.5$) or hip (positive: $p = 0.9$, negative: $p = 0.1$).

### 3.4. Spatial-temporal gait parmeters

There was no significant difference in the stride frequency ($p = 0.09$), step width ($p = 0.1$), or step length ($p = 0.9$) between conditions. In addition, there was no significant difference in step width variability ($p = 0.6$) or step length variability ($p = 0.2$).

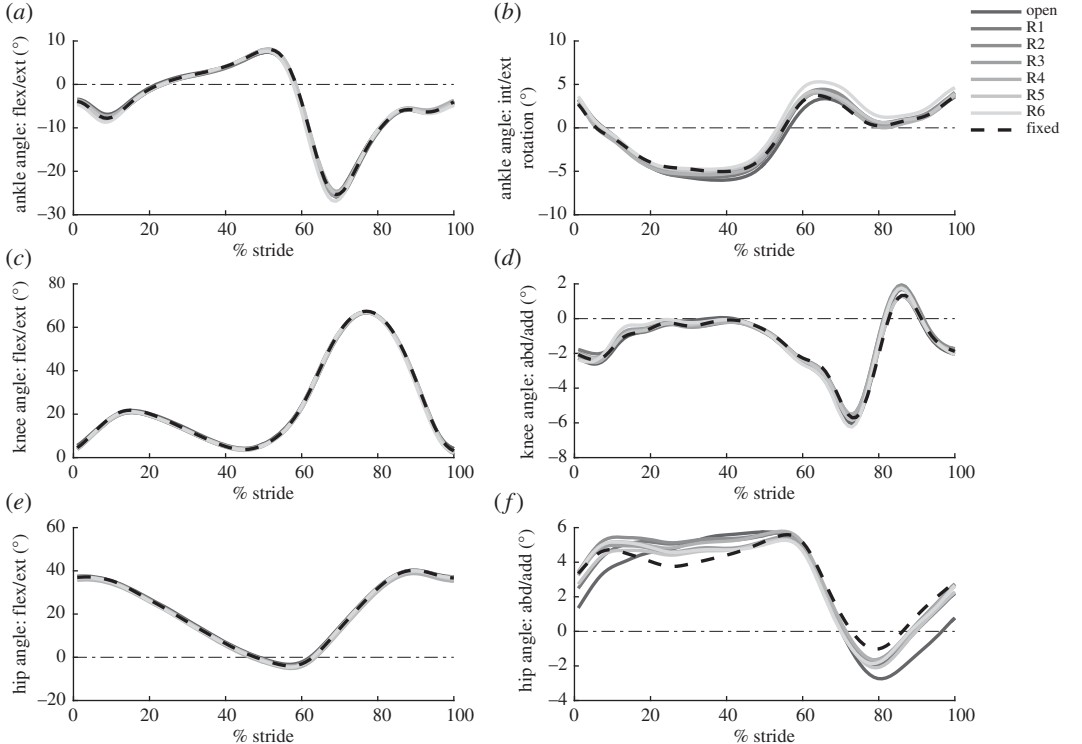

**Figure 6.** Joint angles of the ankle, knee and hip in the sagittal and frontal plane. (*a*) The sagittal plane ankle angle as a function of % gait cycle. Each waveform represents the average across all subjects for that condition. Increased line transparency indicates an increase in damping from energy harvesting. (*b*) The frontal plane ankle angle as a function of % gait cycle. (*c*) The sagittal plane knee angle as a function of % gait cycle. (*d*) The frontal plane knee angle as a function of % gait cycle. (*e*) The sagittal plane hip angle as a function of % gait cycle. (*f*) The frontal plane hip angle as a function of % gait cycle.

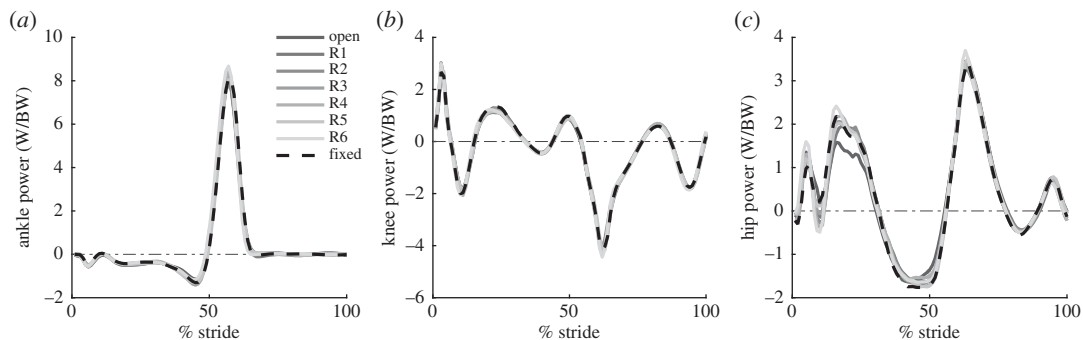

**Figure 7.** Joint power of the ankle, knee and hip. (*a*) The joint power of the ankle as a function of % gait cycle. Each waveform represents the average across all subjects for that condition. Increased line transparency indicates an increase in damping from energy harvesting. (*b*) Knee joint power as a function of % gait cycle. (*c*) Hip joint power as a function of % gait cycle.

## 4. Discussion

In this study, we tested a novel energy harvesting load carriage device for use during walking. The load carriage device was able to generate up to $0.22 \pm 0.03$ W of electricity while walking with 9 kg of weight, with no significant changes in the metabolic power required to walk. These results indicate that our approach to extracting mechanical power from human motion, and converting it to electrical power, results in no significant metabolic penalty to the user during the load carriage task. This has significant implications for the design of load carriage devices as we are the first to show that a load carriage device that allows carried weight to oscillate in the M–L direction can reduce peak M–L forces on the user by an average $27 \pm 4\%$, generate electricity, and result in no significant metabolic power increase for the user.

The advantage of the M−L oscillating load carriage device is that it is capable of harvesting electricity without dedicated attention to its use, unlike traditional hand crank or foot pedal generators. The $0.22 \pm 0.03$ W of electricity generated by our device is sufficient to power smaller portable electronics such as satellite beacons used to send distress signals (∼0.006 W — Spot Gen3) [26] or global positioning systems (GPS) used for navigation (∼0.06 W—Garmin inReach SE) [27]. Modelling predicts that an increase in electrical power production could be achieved by increasing the weight carried. If generating over 1 W of electrical power was desired for powering higher demand devices, such as talking or browsing the internet with a cellphone (∼0.6 − 1 W—Samsung Galaxy S8) [28], our model indicates that over 20 kg of weight would need to be carried in the device. As subjects walked with an average metabolic power of $296 \pm 22$ W during the R3 condition, harvesting an 0.8 W of additional electricity from the user should still have a negligible impact on the overall energetics of locomotion.

Assuming a lithium-ion battery's specific energy is around 720 KJ kg$^{-1}$ [4], a battery with an equivalent weight of our device would contain 1584 KJ of energy. Under the conditions tested in this study (carrying 9 kg and producing 0.22 W), a subject would need to walk 2000 h to generate the same amount of electrical energy. For the energy harvesting device to become a viable alternative to carrying batteries, the weight of the device must be reduced in future iterations. For example, if instead, a subject walked with a configuration where 20 kg was in their backpack and 1 W was being generated by the device (as predicted by our device model). The device would need to weigh as little as 0.6 kg to be the superior choice over batteries for a two week hike (assuming 8 h of walking a day). For the device to be considered as an alternative to batteries, future versions of the device should consider reducing the weight or increasing the efficiency of the device.

Damping from energy harvesting was proposed as a viable tool to control mass oscillation dynamics. This new dimension in device control was used in this study to determine optimal device behaviour, resulting in favourable device−user interaction. Mass oscillation amplitude was identified in our passive M−L load carriage evaluation study [18] to be a possible way to reduce the frontal plane interaction moment, where the frontal plane interaction moment was thought to be a potential reason for the increase in metabolic power when walking with the passive version of the device. By changing the amount of damping present, we were able to effectively control mass oscillation amplitude and decrease the frontal plane interaction moment, demonstrating the device's capabilities to exert varying device−user interaction dynamics. The greatest reduction in frontal plane interaction moment occurred in the R3 condition, where the moment was reduced by 41% compared to the open condition. As hypothesized, this also coincided with the only condition that exhibited a significant decrease in metabolic cost compared to the open condition.

The metabolic power required to walk with the device was greatest during the open condition and the smallest during the R3 condition. There was a significant reduction in metabolic power between the open and R3 condition. Traditional paradigms of joint work and power did not explain the difference in metabolic cost, as there was no significant differences in peak power or work done by any lower limb joint. However, there were three changes in how the device interacted with the user that may explain the changes observed in metabolic power. First, we observed a reduction in the phase angle of carried mass oscillations, with respect to trunk M−L excursions, when damping was increased. Studies that have looked at vertical oscillating load carriage [29–31] have suggested that entirely out-of-phase oscillations, with reduced damping, decrease mechanical work over a stride and the metabolic cost of walking. Either the opposite relationship is true for M−L load carriage, or the phase angle of oscillations had a negligible impact compared to other factors. The second change in backpack-user interaction we observed was an increase in the peak M−L interaction force with increased damping. Although no direct comparisons are known to the authors, Ijmker [21] found that metabolic power decreased with increased M−L stabilization force. However, the M−L forces subject's experienced in [21] were completely out-of-phase with trunk excursions, unlike the interaction forces users experienced with our device. As well, the difference in M−L interaction force between the open and R6 condition is relatively small (less than 2 N). Lastly, one possible reason for why we observed this particular metabolic power difference is the device frontal plane interaction moment. Similar to the metabolic cost, the frontal plane moment is greatest during the open condition, and smallest during the R3 condition. Any moment developed by the load carriage device would be transferred to the subject's trunk and pelvis, through the backpack straps and frame, and counteracted by the trunk and lower-limbs. Such a counteraction would require activation of musculature, in particular the trunk, which could lead to an increase in the metabolic power of walking. Alternatively, the frontal plane interaction moment could simply be correlated with metabolic power, instead of being the cause.

Another possible reason for the observed metabolic landscape is perhaps due to muscle co-contraction. With no damping, the mass undergoes large deviations as a result of non-steady-state trunk input. That is, non-steady-state trunk behaviour that arises due to imperfect motor control and variable step placement, can result in an increase in pendulum oscillation amplitude and therefore energy stored in the inverted pendulum. With damping, these increased oscillations of the carried mass are attenuated back to steady-state. Perhaps as a feedforward mechanism in anticipation of large unattenuated mass oscillations [32], there is an increase in co-contraction of abdominal or hip abb/abduction musculature due to perceived threat in stability [33]: increasing the body's ability to act like a damper and removing additional energy from the oscillating pendulum, but at a cost of additional muscle activation and metabolic power. Lastly, we observed a significant increase in the range-of-motion of the hip in the frontal plane during the open condition, compared to fixed. However, this difference was relatively small, and there was no significant difference in hip range-of-motion between the R3 condition and open.

During testing, although we continued to increase the estimated damping of the system, the observed reduction in amplitude ratio and phase angle became less. This, we suspect, is due to a reduction in the actual damping delivered to the pendulum, compared to the estimated damping predicted by our model. The decrease in effective damping is thought to be primarily due to backlash within the device's gear train causing periods of zero generator torque transmission after reversal of pendulum velocity direction. At lower oscillation amplitudes, the pendulum oscillates primarily within the slack created by backlash: therefore, transmitting little rotation to the generator. Since increased damping reduces the oscillation amplitude, higher damping conditions are more affected by backlash. This is thought to be why we observed the relationship of amplitude ratio and phase angle with respect to damping. This is also thought to be why we observed a shift in peak power production to the R5 condition, instead of the R2 condition. The R2 condition was predicted by the model to be the best balance between harvesting the most electricity from pendulum oscillations, without damping the pendulum's oscillations to a suboptimal point. However, since the effective damping achieved was less than what was modelled, a condition with higher modelled damping (R5) ended up producing the most electrical power. Future modifications, such as higher precision gears and assembly, scissor gears, or using mechanical rectification with one-way clutching and two drive trains could reduce backlash within the system. However, reducing backlash may not increase the maximum electrical power production. This is because modelling has shown that electrical power production decreases with increased damping past the optimal damped condition. Therefore, using a system with no mechanical play, that is capable of higher levels of damping than the current system, may not lead to higher electrical power production.

As with the previous experiment [18] using the load-carriage device (with no energy harvesting module), we observed a reduction in the M−L interaction force when the mass was oscillating, compared to fixed. However, we did not observe in this current study a reduction in vertical interaction force as we had with the previous study. There are methodological differences between the two studies, such as amount of weight carried and free versus prescribed step width, that may contribute to this discrepancy. The greatest difference, however, may be within the device itself. The closest parallel to the previous iteration of the device is the current device operating under the open condition. However, there is still a significant increase in the mechanical damping present with the energy harvesting module in an open circuit condition, compared to the previous device without the energy harvesting module. This additional damping causes a shift in phase and amplitude of mass oscillations, and therefore may create different device–user interaction forces. This may explain the differences in load carriage device results observed in this study, compared to the previous.

Methodological changes between the two studies may have also caused differences in walking behaviour observed in each study. In the previous experiment, subjects walked at prescribed step widths greater, less, and at the preferred step width determined during an acclimatization session the day before, when subjects walked with the carried mass in a fixed condition. This is opposed to the current study where step width was freely chosen during each condition. This methodological difference could be why we observed several differences in gait-related measures between the two studies. First, we observed kinematic changes in the frontal plane hip angle that were not observed in the previous study. Also, subjects chose to walk at a more narrow step width under the oscillating condition in the previous study, when given the opportunity to freely choose step width. As we did not observe this behaviour in this study, we suspect that perhaps because the preferred step width during the fixed condition was determined the day before, during an acclimatization session, that subjects may have decreased their preferred step width after more experience walking in the

laboratory with the device. Lastly, in the current study, we observed no significant changes in hip work done when walking with an oscillating mass, where as we observed an increase in hip work performed in the previous study. In the previous study, however, increases in hip work increased at wider prescribed step widths, notably near and greater than the preferred step width. Therefore, when subjects walked at a preferred step width in this study, their step width may not have been wide enough to elicit significant changes in hip work between oscillating and fixed conditions.

A limitation to the following study was tracking the pelvis segment using the trunk of the subject. Although a limitation, we expect that this method affects all conditions equally. However, caution should be exercised, in particular with hip joint angle calculations.

# 5. Conclusion

In summary, we developed a novel energy harvesting load carriage device that generated electricity, reduced peak interaction forces experienced by the user, and resulted in no significant increase in metabolic power to operate, compared to a fixed mass. We used the energy harvesting module to control device behaviour in order explore optimal load carriage device–user interactions. Our novel approach to use damping as a method to explore amplitude of carried mass oscillations allowed us to tune device behaviour such that we achieved electrical power generation with no significant increase in metabolic cost. Future work should consider reducing the mass of the energy harvesting backpack system to reduce carrying costs associated with walking with the device. Overall, these results help contribute to the understanding of load carriage interaction forces and moments and their implications for gait biomechanics.

Ethics. All study participants provided written informed consent prior to the experimentation. The experiment was approved by the General Research Ethics Board of Queen's University.

Data accessibility. All data and analysis code used in this manuscript can be found at Dryad at: http://dx.doi.org/10.5061/dryad.q38417c [34].

Authors' contributions. J.P.M. fabricated the device, conducted the experiments, analysed the results and wrote the manuscript. Q.L. conceived the device and supervised the project. All authors designed the device, conceived the experiment and reviewed the manuscript.

Competing interests. The authors declare no competing interests.

Funding. This work was supported by Natural Sciences and Engineering Research Council Discovery Grant awarded to Q.L. and the Alexander Graham Bell Scholarship awarded to J.P.M.

Acknowledgements. Thanks to Laura Hutchinson for assistance in data collection. Thanks to Dr Ronald Anderson for his input on device modelling and design. Thanks to Dr Kevin Deluzio for providing the Human Mobility Research Laboratory.

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
