## [Reviewer comments · Royal Society Open Science]

Review History

RSOS-182021.R0 (Original submission)

Review form: Reviewer 1

Is the manuscript scientifically sound in its present form?

Yes

Are the interpretations and conclusions justified by the results?

Yes

Is the language acceptable?

Yes

Is it clear how to access all supporting data?

Yes

Do you have any ethical concerns with this paper?

No

Have you any concerns about statistical analyses in this paper?

No

Recommendation?

Accept with minor revision (please list in comments)

Comments to the Author(s)

This is an interesting paper which explores how a medial-lateral oscillating load may affect the energetics and biomechanics of human walking. I suggest addressing the following discussion points prior to publication:

Minor Revisions/Discussion Points:

- Why was a 67:1 gear ratio chosen? What makes this gear ratio ideal for the Maxxon motor to generate electricity at max efficiency?
- Would more power have been generated if the gear backlash issue could be solved? Would spring-loaded anti-backlash gears help prevent the backlash issue, or would some other method be useful? Could the effective damping due to gear backlash have affected any other results beside power generation?
- On line 284-285, it is stated that the "... carried weight to oscillate in the M-L direction can reduce peak forces on the user by an average 27.4%, generate electricity, ..." This statement is slightly misleading as the vertical peak forces were not affected and are ~10x larger in magnitude than the M-L forces. It should be noted here that the peak M-L forces were reduced by an average of 27.4%, not the peak forces on the user generally.
- Prior work with suspended load backpacks suggests that the ideal configuration may be when the load is completely out-of-phase with the body. The phase angle of the load in this paper was 130 +/- 20.8 degrees which decreased with higher damping. Would approaching a 180 degree phase shift be ideal for maximum energy harvesting, or is some damping necessary for this design? Why was R2 predicted to be the ideal damping ratio for maximum energy harvesting?
- Though it's impressive that power can be generated from M-L oscillation of a backpack without significant effects on the energetics and biomechanics of walking, can you put this power generation in context? The electricity generation was only 0.22 +/- 0.3 W. A smartphone charges at 5V and ~1-2 amps, so about 10-20 W. This device would be able to trickle-charge a smartphone in remote areas. Is this technology worthwhile considering the added backpack weight and cost, especially considering other comparable methods of converting mechanical work into electricity (i.e. hand crank, footpedal, etc.)? Could more energy be harvested with modifications to your system? What is the theoretical maximum energy that could be harvested with a M-L oscillating load? Also, human walking requires roughly 30-75 W of mechanical power, so it would make sense that harvesting 0.22 W at a reasonable efficiency may not significantly affect the mechanical or metabolic cost of walking. It would be helpful to put the power output in context.

Review form: Reviewer 2**Is the manuscript scientifically sound in its present form?**

Yes

Are the interpretations and conclusions justified by the results?

Yes

Is the language acceptable?

Yes

Is it clear how to access all supporting data?

No

Do you have any ethical concerns with this paper?

No

Have you any concerns about statistical analyses in this paper?

Yes

Recommendation?

Accept with minor revision (please list in comments)

Comments to the Author(s)

Attached (Appendix A).

Decision letter (RSOS-182021.R0)

28-Mar-2019

Dear Mr Martin,

The editors assigned to your paper ("Generating electricity while walking with a medial-lateral oscillating load carriage device") have now received comments from reviewers. We would like you to revise your paper in accordance with the referee and Associate Editor suggestions which can be found below (not including confidential reports to the Editor). Please note this decision does not guarantee eventual acceptance.

Please submit a copy of your revised paper before 20-Apr-2019. Please note that the revision deadline will expire at 00.00am on this date. If we do not hear from you within this time then it will be assumed that the paper has been withdrawn. In exceptional circumstances, extensions may be possible if agreed with the Editorial Office in advance. We do not allow multiple rounds of revision so we urge you to make every effort to fully address all of the comments at this stage. If deemed necessary by the Editors, your manuscript will be sent back to one or more of the original reviewers for assessment. If the original reviewers are not available, we may invite new reviewers.

- Data accessibility

If you wish to submit your supporting data or code to Dryad (<http://datadryad.org/>), or modify your current submission to dryad, please use the following link:
<http://datadryad.org/submit?journalID=RSOS&manu=RSOS-182021>

- Competing interests

- Authors' contributions

- Acknowledgements

- Funding statement

on behalf of Dr Monica Daley (Associate Editor) and R. Kerry Rowe (Subject Editor)
 openscience@royalsociety.org

Associate Editor's comments (Dr Monica Daley):

Associate Editor: 1

Comments to the Author:

Two experts have reviewed your paper and they are generally positive about the approach and contribution. They do raise a number of questions for the authors to address in a revised version, clarifying aspects of the device design and function, the experimental measures and the interpretation of the energy harvesting capacity of the device relative to alternative approaches. I would like to invite the authors to submit a revised version of the manuscript that addresses the points raised. Please ensure that the resubmission includes a point-by-point response to the referees, and please also ensure that the data and code are made available in an archival data repository in line with RSOS open data policies.

Associate Editor: 2

Comments to the Author:
 (There are no comments.)

Comments to Author:

Reviewers' Comments to Author:

Reviewer: 1

Comments to the Author(s)

This is an interesting paper which explores how a medial-lateral oscillating load may affect the energetics and biomechanics of human walking. I suggest addressing the following discussion points prior to publication:

Minor Revisions/Discussion Points:

- Why was a 67:1 gear ratio chosen? What makes this gear ratio ideal for the Maxxon motor to generate electricity at max efficiency?
- Would more power have been generated if the gear backlash issue could be solved? Would spring-loaded anti-backlash gears help prevent the backlash issue, or would some other method be useful? Could the effective damping due to gear backlash have affected any other results beside power generation?
- On line 284-285, it is stated that the "... carried weight to oscillate in the M-L direction can reduce peak forces on the user by an average 27.4%, generate electricity, ..." This statement is slightly misleading as the vertical peak forces were not affected and are ~10x larger in magnitude than the M-L forces. It should be noted here that the peak M-L forces were reduced by an average of 27.4%, not the peak forces on the user generally.

- Prior work with suspended load backpacks suggests that the ideal configuration may be when the load is completely out-of-phase with the body. The phase angle of the load in this paper was 130 ± 20.8 degrees which decreased with higher damping. Would approaching a 180 degree phase shift be ideal for maximum energy harvesting, or is some damping necessary for this design? Why was R2 predicted to be the ideal damping ratio for maximum energy harvesting?
- Though it's impressive that power can be generated from M-L oscillation of a backpack without significant effects on the energetics and biomechanics of walking, can you put this power generation in context? The electricity generation was only 0.22 ± 0.3 W. A smartphone charges at 5V and $\sim 1-2$ amps, so about 10-20 W. This device would be able to trickle-charge a smartphone in remote areas. Is this technology worthwhile considering the added backpack weight and cost, especially considering other comparable methods of converting mechanical work into electricity (i.e. hand crank, footpedal, etc.)? Could more energy be harvested with modifications to your system? What is the theoretical maximum energy that could be harvested with a M-L oscillating load? Also, human walking requires roughly 30-75 W of mechanical power, so it would make sense that harvesting 0.22 W at a reasonable efficiency may not significantly affect the mechanical or metabolic cost of walking. It would be helpful to put the power output in context.

Reviewer: 2

Comments to the Author(s)
Attached

Author's Response to Decision Letter for (RSOS-182021.R0)

See Appendices B & C.

Decision letter (RSOS-182021.R1)

24-Apr-2019

Dear Mr Martin:

On behalf of the Editors, I am pleased to inform you that your Manuscript RSOS-182021.R1 entitled "Generating electricity while walking with a medial-lateral oscillating load carriage device" has been accepted for publication in Royal Society Open Science subject to minor revision in accordance with the referee suggestions. Please find the referees' comments at the end of this email.

The reviewers and Subject Editor have recommended publication, but also suggest some minor revisions to your manuscript. Therefore, I invite you to respond to the comments and revise your manuscript.

- Ethics statement

- Data accessibility

If you wish to submit your supporting data or code to Dryad (<http://datadryad.org/>), or modify your current submission to dryad, please use the following link:
<http://datadryad.org/submit?journalID=RSOS&manu=RSOS-182021.R1>

- Competing interests

- Authors' contributions

- Acknowledgements

- Funding statement

Because the schedule for publication is very tight, it is a condition of publication that you submit the revised version of your manuscript before 03-May-2019. Please note that the revision deadline will expire at 00.00am on this date. If you do not think you will be able to meet this date please let me know immediately.

on behalf of Dr Monica Daley (Associate Editor) and R. Kerry Rowe (Subject Editor)
openscience@royalsociety.org

Associate Editor Comments to Author (Dr Monica Daley):

Associate Editor

Comments to the Author:

Thank you for thoroughly addressing the reviewers comments on the earlier version of your manuscript. I am happy to accept the paper for publication in RSOS, subject to compliance with the open data policy. I can see that provided google drive link provides the data as requested. However, a google drive link does not create a permanent and citable database. The code and data should be provided in a curated public database such as Dryad or Figshare, which will also allow the data to be linked to the DOI of the paper. Please see details here: <https://royalsociety.org/journals/ethics-policies/data-sharing-mining/>

Author's Response to Decision Letter for (RSOS-182021.R1)

See Appendix D.

Decision letter (RSOS-182021.R2)

08-May-2019

Dear Mr Martin,

I am pleased to inform you that your manuscript entitled "Generating electricity while walking with a medial-lateral oscillating load carriage device" is now accepted for publication in Royal Society Open Science.

on behalf of Dr Monica Daley (Associate Editor) and R. Kerry Rowe (Subject Editor)
openscience@royalsociety.org

Associate Editor Comments to Author (Dr Monica Daley):

Associate Editor: 1

Comments to the Author:

(There are no comments.)

Reviewer comments to Author:

Appendix A

Title: Generating electricity while walking with a medial-lateral oscillating load carriage device.

The purpose of this study was to design and test a biomechanical energy harvesting backpack that utilizes movement in the medial-lateral direction. In this study the interaction between the electromagnetic damping on a medial-lateral oscillating load carriage, walking kinematics, and kinetic and electrical power generation are investigated.

This paper is well written and the experiments seems to be well conceived and performed.

I have few comments that I think would improve the paper and its impact.

- Main issues:

1. Metabolic rate was calculated over only the last 2 min.

Our experience and that of others has shown that because of the variability in these measurements 2 min are not sufficient (3-4 min are preferable). Can the authors start use the data from earlier time? Typically, in our experiments subjects reach steady-state state after 1.5 min, and we start after 3min, which is most common in the literature.)

2. In the conclusions, the authors talk about the effect of the additional mass but do not quantify it. I think it is important to do so.

Can the authors calculate what is the required mass of the device to become worth using? As a criterion they could use the change in metabolic rate due to addition of the device mass and compared it to the metabolic rate saved using the device vs carrying it as a fix load.

Options for equations that calculate change in metabolic rate as a function of the add mass and walking speed

Pandolf, K. B., Givoni, B. and Goldman, R. F. (1977). Predicting energy expenditure with loads while standing or walking very slowly. *J. Appl. Physiol.* 43, 577-581,

Givoni, B. and Goldman, R. F. (1971). Predicting Metabolic Energy Cost. *J. Appl. Physiol.* 30, 429-433.

Scherzter E, Riemer R. Metabolic rate of carrying added mass: a function of walking speed, carried mass and mass location. *Appl Ergonom.* 2014;45:1422-32.

Or another possibility as an alternative for energy harvesting

Scherzter, E. and Riemer, R. 2015. Harvesting biomechanical energy or carrying batteries? An evaluation method based on a comparison of metabolic power. *Journal of NeuroEngineering and Rehabilitation*, Vol 12, Issue 30.

3. Discussion

In the paper, four components describe the interaction of the device with the human (the system frequency, the phase shift, the m-l force, and the frontal plane moment). Yet, the discussion fails to provide insight into the role of each of these components and the interaction between them. There are no references to papers that might help in

providing such an insight. Furthermore, even if such papers are not available, I think you could do it graphically or by using statistical modeling (e.g., linear mixed models with these factors explaining the change in metabolic rate)

- Specific Comments

The link for your data and code did not work. So perhaps some of the comments I have made are, in fact, addressed in these files.

Line 88-103

The masses of the parts and the total mass should be in table in the paper or in the supplementary information.

Line 89

It is hard to understand from this figure how this device works; perhaps the authors can add a side view. Conceptually the figure should look something like this:

Line 105

Again, the link for your data and code did not work. So perhaps this information is there but if not, it would be a good idea to give the data and performance of each of the subjects.

Line 124

The out of phase analysis is worth a figure (if not in the main paper then at least in the supplementary information).

Line 127

How did the authors apply an equivalent damping ratio of 0.7?

Line 132-133

How does the damping ratio translate to the system frequency? Like this:?

$$W_d = \sqrt{1 - \zeta^2}$$

(if so, I recommend putting in the equation).

Also, did the stepping frequency change in the different conditions?

There seems to be a difference in the reported sampling frequency (200 vs 100) of the motion data in two places is it the same data?

Line 142, 182

Line 142

“**200** Hz and filtered using a 2nd order low-pass zero-phase Butterworth”

Line 182

100Hz , also here you don't report the cut off frequency?

Line 150-152

Again, it is difficult to understand from the text what the authors are comparing. Are both motions are linear or angular?

Line 156 -163

It will be good if the authors could add to figure 1 the force sensor location and how they mounted the sensor to measure both the force and moments.

Line 191-192

Should be joint kinematics were calculated using XXX [e.g. bottom-up approach (Winter 2009)], and segment mass center and moment of inertia where calculated using [25]

Line 214

Why do the authors think that R2 condition would be the best? I think they should let the readers know.

Line 244-55

Was there difference between the different harvesting conditions?

Line 258

In the figure you write Abd/add in the text you use frontal plane, I know you mean the same. Yet for clarity you might want to write something like Abd/add (frontal plane) or use the same wording in both places.

Line 338-340

Not clear how the peak power was affected by the backlash.

Line 370-373

How did the step width change with the different harvesting conditions and how is this related (if at all) to the frontal torque?

Appendix B

Reviewer 1: Author Response

We thank the reviewer for their time reviewing the manuscript and their in-depth feedback. We believe the comments they gave helped strengthen the clarity and significance of our results. We believe we have addressed the comments brought up by the reviewer, and by doing so, have strengthened the quality of the manuscript.

Response to the reviewer comments (in order of appearance):

Comments to the Author(s):

This is an interesting paper which explores how a medial-lateral oscillating load may affect the energetics and biomechanics of human walking. I suggest addressing the following discussion points prior to publication:

Minor Revisions/Discussion Points:

Why was a 67:1 gear ratio chosen? What makes this gear ratio ideal for the Maxxon motor to generate electricity at max efficiency?

- *From modelling done on the device, we found that a higher gear ratio results in a higher electrical efficiency. With higher input angular velocity to the generator, the external load resistance can be increased while still delivering the same electromotive torque on the pendulum. This means there is less line current, resulting in a decrease in losses due to the generator's armature resistance. We chose gears, from what was commercially available, that maximized our overall gear ratio while still meeting size and strength constraints. We are also writing a paper on the design, model, and model performance of the device in another paper. To keep the discussion mostly on the effects of the user in this paper, and limit the amount of discussion on the device model, we added a more concise explanation on lines 100-104. It now reads: "An overall gear ratio of 67:1 was chosen to maximize the gear ratio, given the largest commercially available polyacetal gears that also met our space and strength requirements (Misumi USA, IL). A greater gear ratio was determined from modelling to minimize losses due to rotor winding resistance."*

Would more power have been generated if the gear backlash issue could be solved? Would spring-loaded anti-backlash gears help prevent the backlash issue, or would some other method be useful? Could the effective damping due to gear backlash have affected any other results beside power generation?

- *To address your first two comments, we have added the following lines to the Discussion Section: "Future modifications, such as higher precision gears and assembly, scissor gears, or using mechanical rectification with one-way clutching and two drive trains could reduce backlash within the system. However, reducing backlash may not increase the maximum electrical power production. This is because modelling has shown that electrical power production decreases with increased damping past the optimal damped condition. Therefore, given a system with no mechanical play, that is capable of higher levels of damping than the current system, may not lead to higher electrical power production." (Lines 400-406). To address your last point, we do think that backlash did*

affect other performance measures of the device. In particular, we mention how backlash had affected the amplitude ratio and phase angle on Lines 391-394.

On line 284-285, it is stated that the "... carried weight to oscillate in the M-L direction can reduce peak forces on the user by an average 27.4%, generate electricity, ..." This statement is slightly misleading as the vertical peak forces were not affected and are ~10x larger in magnitude than the M-L forces. It should be noted here that the peak M-L forces were reduced by an average of 27.4%, not the peak forces on the user generally.

- *As suggested, we included 'medial-lateral forces' on Line 302 to clarify that reductions were only observed in the medial-lateral direction.*

Prior work with suspended load backpacks suggests that the ideal configuration may be when the load is completely out-of-phase with the body. The phase angle of the load in this paper was 130 +/- 20.8 degrees which decreased with higher damping. Would approaching a 180 degree phase shift be ideal for maximum energy harvesting, or is some damping necessary for this design? Why was R2 predicted to be the ideal damping ratio for maximum energy harvesting?

- *When harvesting electricity using the generator, the generator will apply a torque back on to the pendulum which will dampen the oscillations. Therefore, to generate electricity, there must be damping. However, although more damping means more electricity being harvested by the generator, it also decreases the oscillation amplitude of the pendulum. Therefore, the optimal electrical power condition is predicted to be a balance between damping the oscillations to generate electricity, but not so much as to attenuate the amplitude of oscillations to a suboptimal level.*
- *Based on the reviewers comment we attempt to clarify this point in the Methods section on lines 107-113: "By reducing the value of the electrical resistors in the circuit, the generator harvests more electricity from the angular velocity of the pendulum. However, this also applies more back electromotive torque on the pendulum, which increases the overall damping of the system and leads to decreased mass oscillations. Therefore, maximum electrical power harvested during walking is a balance between harvesting as much electricity from the oscillations of the pendulum without damping the pendulum's oscillations to a suboptimal point."*

Though it's impressive that power can be generated from M-L oscillation of a backpack without significant effects on the energetics and biomechanics of walking, can you put this power generation in context? The electricity generation was only 0.22 +/- 0.3 W. A smartphone charges at 5V and ~1-2 amps, so about 10-20 W. This device would be able to trickle-charge a smartphone in remote areas. Is this technology worthwhile considering the added backpack weight and cost, especially considering other comparable methods of converting mechanical work into electricity (i.e. hand crank, footpedal, etc.)? Could more energy be harvested with modifications to your system? What is the theoretical maximum energy that could be harvested with a M-L oscillating load? Also, human walking requires roughly 30-75 W of mechanical power, so it would make sense that harvesting 0.22 W at a reasonable efficiency may not

significantly affect the mechanical or metabolic cost of walking. It would be helpful to put the power output in context.

- *As suggested by the reviewer, we aim to contextualize the power generated by the device in the Discussion Section on lines 305-317: “The advantage of the medial-lateral oscillating load carriage device is that it is capable of harvesting electricity without dedicated attention to its use, unlike traditional hand crank or foot pedal generators. The 0.22±0.03 W of electricity generated by our device is sufficient to power smaller portable electronics such as satellite beacons used to send distress signals (~0.006 W - Spot Gen3)[26] or global positioning systems (GPS) used for navigation (~0.06 W - Garmin inReach SE)[27]. Modelling predicts that an increase in electrical power production could be achieved by increasing the weight carried. If generating over 1 W of electrical power was desired for powering higher demand devices, such as talking or browsing the internet with a cellphone (~0.6-1 W - Samsung Galaxy S8)[28], our model indicates that over 20 kg of weight would need to be carried in the device. As subjects walked with an average metabolic power of 296±22 W during the R3 condition, harvesting an 0.8 W of additional electricity from the user should still have a negligible impact on the overall energetics of locomotion.”*

Appendix C

Reviewer 2: Author Response

We thank the reviewer for their time reviewing the manuscript and their in-depth feedback. We believe the comments they gave helped strengthen the analysis, interpretation, and clarity of our results. We believe we have addressed the comments brought up by the reviewer, and by doing so, have strengthened the quality of the manuscript.

Response to the reviewer comments (in order of appearance):

The purpose of this study was to design and test a biomechanical energy harvesting backpack that utilizes movement in the medial-lateral direction. In this study the interaction between the electromagnetic damping on a medial-lateral oscillating load carriage, walking kinematics, and kinetic and electrical power generation are investigated.

This paper is well written and the experiments seems to be well conceived and performed.

I have few comments that I think would improve the paper and its impact.

Main Issues:

1. Metabolic rate was calculated over only the last 2 min. Our experience and that of others has shown that because of the variability in these measurements 2 min are not sufficient (3-4 min are preferable). Can the authors start use the data from earlier time? Typically, in our experiments subjects reach steady-state state after 1.5 min, and we start after 3min, which is most common in the literature.)

- *As suggested by the reviewer, we re-processed all metabolic data using the last 3 minutes of the trial. However, we found that the metabolic power across conditions changed on average by only $-0.17\pm 0.40\%$ by incorporating the extra 1 minute of data. Also, incorporating the extra minute of metabolic data did not change conclusions drawn (only significant difference is still Open to R3 condition, $p = 0.04$). Therefore, to keep the analysis period of metabolic power the same as other gait measures (such as kinematics and kinetics) we decided to keep the previous analysis period of 2 minutes.*
- *To support our decision, I have compiled a quick list of 10 studies that use a similar analysis period for metabolic power:*
 - *Ding et al., Human-in-the-loop optimization of hip assistance with a soft exosuit during walking. Science Robotics (2018). Steady state trials were 5 min, averaged last 2 min.*
 - *Wu and Kuo, Determinants of preferred ground clearance during swing phase of human walking. Journal of Experimental Biology (2016). 6 min trials, averaged last 2 min.*
 - *Malcolm et al., Continuous sweep versus discrete step protocols for studying*

effects of wearable robot assistance magnitude. Journal of Neuroengineering and Rehabilitation (2017). 5 min trials, averaged last 2 min.

- *Sanchez et al., Evidence of energetic optimization during adaptation differs for metabolic, mechanical, and perceptual estimates of energetics cost. Scientific Reports (2017). 5 min trials, averaged last 2 min.*
- *Galle et al., Exoskeleton plantarflexion assistance for elderly. Gait and Posture (2017). 5 min trials, averaged last 2 min.*
- *Takahashi et al., Adding stiffness to the foot modulates soleus force-velocity behavior during human walking. Scientific Reports (2016). 7 min trials, averaged last 2 min.*
- *Ijmker et al., Postural threat during walking: effects on energy cost and accompanying gait changes. Journal of Neuroengineering and Rehabilitation (2014). 5 min trials, averaged last 2 min.*
- *Quinlivan et al., Assistance magnitude versus metabolic cost reductions for a tethered multiarticular soft exosuit. Science Robotics (2017). 5 min trials, averaged last 2 min.*
- *Mooney et al., Autonomous exoskeleton reduces metabolic cost of human walking during load carriage. Journal of Neuroengineering and Rehabilitation (2014). Variable trial length, averaged last 2 min.*
- *Kim and Bertram, Compliant walking appears metabolically advantageous at extreme step lengths. Gait and Posture (2018). 5 min trials, averaged last 2 min.*

2. In the conclusions, the authors talk about the effect of the additional mass but do not quantify it. I think it is important to do so. Can the authors calculate what is the required mass of the device to become worth using? As a criterion they could use the change in metabolic rate due to addition of the device mass and compared it to the metabolic rate saved using the device vs carrying it as a fix load. Options for equations that calculate change in metabolic rate as a function of the add mass and walking speed:

Pandolf, K. B., Givoni, B. and Goldman, R. F. (1977). Predicting energy expenditure with loads while standing or walking very slowly. *J. Appl. Physiol.* 43, 577-581,

Givoni, B. and Goldman, R. F. (1971). Predicting Metabolic Energy Cost. *J. Appl. Physiol.* 30, 429–433.

Scherzter E, Riemer R. Metabolic rate of carrying added mass: a function of walking speed, carried mass and mass location. *Appl Ergonom.* 2014;45:1422–32.

Or another possibility as an alternative for energy harvesting Scherzter, E. and Riemer, R. 2015.

Harvesting biomechanical energy or carrying batteries? An evaluation method based on a comparison of metabolic power. *Journal of NeuroEngineering and Rehabilitation*, Vol 12, Issue 30.

- *As suggested by the reviewer, a discussion of device weight has been added to the Discussion section (lines 318-328). As the metabolic power saved using the device is small and not statistically significant (compared to the fixed mass condition), we make a comparison of the weight of the device to the weight of batteries only. The following has been added to address this on lines 318-328: "Assuming a lithium-ion battery's specific energy is around 720 KJ/kg [4] a battery with an equivalent weight of our device would contain 1584 KJ of energy. Under the conditions tested in this study (carrying 9 kg and producing 0.22 W), a subject would need to walk 2000 hours to generate the same amount of electrical energy. For the energy harvesting device to become a viable alternative to carrying batteries, the weight of the device must be reduced in future iterations. For example, if instead a subject walked with a configuration where 20 kg was in their backpack and 1 W was being generated by the device (as predicted by our device model). The device would need to weigh as little as 0.6 kg to be the superior choice over batteries for a 2 week hike (assuming 8 hours of walking a day). For the device to be considered as an alternative to batteries, future versions of the device should consider reducing the weight or increasing the efficiency of the device."*

3. Discussion: In the paper, four components describe the interaction of the device with the human (the system frequency, the phase shift, the m-l force, and the frontal plane moment). Yet, the discussion fails to provide insight into the role of each of these components and the interaction between them. There are no references to papers that might help in providing such an insight. Furthermore, even if such papers are not available, I think you could do it graphically or by using statistical modeling (e.g., linear mixed models with these factors explaining the change in metabolic rate)

- *As suggested by the reviewer, we expanded our discussion on changes to the device-user interaction across conditions. First, it is worth noting that the system's natural frequency and the carried mass' oscillation frequency do not change between conditions. The carried mass always oscillates at the same frequency as M-L COM excursions. Therefore, we performed a linear mixed-model statistical analysis using peak M-L force, peak frontal plane moment, and phase angle as fixed effects with interactions, subjects as a random effect, and metabolic power as a response variable. First, from visual inspection there was no obvious linear relationship between fixed effects and the response variable. This was also the case with the linear mixed-model, where neither three fixed effects, or their interactions, were statistically significant.*
- *As an alternative to reporting on a statistical model that has no significant predictive power, we added an additional discussion, using previous literature when possible, on the potential effects of device-user interaction changes on lines 347-361 in the Discussion Section: "However, there were three changes in how the device interacted*

with the user that may explain the changes observed in metabolic power. First, we observed a reduction in the phase angle of carried mass oscillations, with respect to trunk M-L excursions, when damping was increased. Studies that have looked at vertical oscillating load carriage [29-31] have suggested that entirely out-of-phase oscillations, with reduced damping, decrease mechanical work over a stride and the metabolic cost of walking. Either the opposite relationship is true for M-L load carriage, or the phase angle of oscillations had a negligible impact compared to other factors. The second change in backpack-user interaction we observed was an increase in the peak M-L interaction force with increased damping. Although no direct comparisons are known to the authors, Ijmker [21] found that metabolic power decreased with increased M-L stabilization force. However, the M-L forces subject's experienced in [21] were completely out-of-phase with trunk excursions, unlike the interaction forces users experienced with our device. As well, the difference in M-L interaction force between the Open and R6 condition is relatively small (<2 N). Lastly, one possible reason for why we observed this particular metabolic power difference is the device frontal plane interaction moment. Similar to the metabolic cost,...”

Specific Comments :

The link for your data and code did not work. So perhaps some of the comments I have made are, in fact, addressed in these files.

- *As pointed out by the reviewer, the previous URL led to the main Matlab script and not the entire repository. The URL should now be a hyperlink and properly link to the entire database folder.*

Line 88-103: The masses of the parts and the total mass should be in table in the paper or in the supplementary information.

- *As suggested by the reviewer, a table was added (Table 1) to list masses of parts and the total mass of the device.*

Line 89: It is hard to understand from this figure how this device works; perhaps the authors can add a side view. Conceptually the figure should look something like this:

- *As suggested by the reviewer, an additional figure was added to Figure 1 to show a reduced diagram of how the device works. This can now be seen as the top-right blowout in Figure 1. What was previously Figure 1c is now the bottom-right blowout in Figure 1. What was previously Figure 1c has been moved to a new figure, Figure 2.*

Line 105: Again, the link for your data and code did not work. So perhaps this information is there but if not, it would be a good idea to give the data and performance of each of the subjects.

- *As discussed in an earlier comment, the URL for the study's data repository should now function properly. Individual subject information and data can be found there.*

Line 124: The out of phase analysis is worth a figure (if not in the main paper then at least in the supplementary information).

- *As suggested by the reviewer, we included an additional figure (Figure 2b) that demonstrates the out-of-phase oscillations of the carried mass with respect to the input motion of the user. Another manuscript dedicated to modelling device behavior (and the determination of the frequency ratio) has been submitted to another journal. Therefore, we feel this analysis is outside of the scope of this paper, which we intended to be on experimental results of subjects walking with the device alone.*

Line 127: How did the authors apply an equivalent damping ratio of 0.7?

- *As the explanation of how an equivalent damping ratio is calculated isn't discussed until several lines after Line 127 (as noted by the reviewer), the wording in the sentence (now line 138) was changed to: "subjects walked with the highest damped condition used in testing (R6)". Now, after the first mention of the equivalent damping ratio on lines 141-143, an explanation of how it is calculated proceeds it (lines 143-148)".*

Line 132-133: How does the damping ratio translate to the system frequency? Like this:?

$$W_d = \sqrt{1 - \zeta^2}$$

(if so, I recommend putting in the equation).

- *To clarify the definition of damping ratio, we included the following on lines 143-144: “The damping ratio, ζ , is defined as the ratio of the modelled damping of the system to the modelled critical damping for the system.” The damping ratio does relate the damped natural frequency to the undamped natural frequency using the following equation:*

$$\omega_d = \omega_n \sqrt{1 - \zeta^2}$$

However, because the undamped natural frequency has not been mentioned or defined in the manuscript, we feel it is unnecessary to include this relationship. We hope that the sentence added to the manuscript above suffices in defining the damping ratio and its relation to the system dynamics.

Also, did the stepping frequency change in the different conditions?

- *As most our analysis is done with reference to the gait cycle, we calculated the stride frequency instead of the stepping frequency for each subject and condition. The stride frequency did not significantly change between conditions as reported in the Spatial-Temporal Gait Parameters Results Section on line 291.*

There seems to be a difference in the reported sampling frequency (200 vs 100) of the motion data in two places is it the same data?

Line 142, 182

Line 142

“**200** Hz and filtered using a 2nd order low-pass zero-phase Butterworth”

- *As pointed out by the reviewer the sampling frequency was reported in error on second mention in the manuscript (line 182). It has now been corrected to 200 Hz (line 193).*

Line 182: **100**Hz, also here you don't report the cut off frequency?

- *As suggested we now include the cutoff frequency used in the Butterworth filter on lines 193-194.*

Line 150-152: Again, it is difficult to understand from the text what the authors are comparing. Are both motions linear or angular?

- *As suggested, we clarified on lines 161-163 that both motions are the linear M-L displacement of the carried mass and input motion.*

Line 156 -163: It will be good if the authors could add to figure 1 the force sensor location and how they mounted the sensor to measure both the force and moments.

- *As suggested, a side view of the load cell mount between device and backpack frame is*

now shown as the center-right blowout in Figure 1.

Line 191-192: Should be joint kinematics were calculated using XXX [e.g. bottom-up approach (Winter 2009)], and segment mass center and moment of inertia where calculated using [25].

- *The method used to calculate joint kinematics and kinetics is referenced earlier in the paragraph (lines 191-192). Therefore, to address the need for clarity as mentioned by the reviewer, we changed the sentence on lines 203-204 to: "Segment mass centre and the moment of inertia of limb segments were determined using [25]."*

Line 214: Why do the authors think that R2 condition would be the best? I think they should let the readers know.

- *As suggested, we first clarify in the Methods Section the underlying principles for why a condition may generate more electricity than another (lines 107-113): "By reducing the value of the electrical resistors in the circuit, the generator harvests more electricity from the angular velocity of the pendulum. However, this also applies more back electromotive torque on the pendulum, which increases the overall damping of the system and leads to decreased mass oscillations. Therefore, maximum electrical power harvested during walking is a balance between harvesting as much electricity from the oscillations of the pendulum without damping the pendulum's oscillations to a suboptimal point."*
- *Then, we re-iterate this point in the Results Section as suggested by the reviewer (lines 226-229): "This suggests that the balance between harvesting the most electricity from pendulum oscillations, without damping the pendulum's oscillations to a suboptimal point, occurs at higher damping ratios than what was predicted by the model."*

Line 244-55: Was there difference between the different harvesting conditions?

- *As suggested by the reviewer, we also report the significant differences between oscillating conditions on lines 258-260: "There was a significant decrease in the peak moment during the R3 condition compared to the Open ($p = 0.04$) and R1 ($p = 0.04$) condition. As well, there was a significant decrease in the peak moment in the R2 condition compared to the Open condition ($p = 0.04$)."*

Line 258: In the figure you write Abd/add in the text you use frontal plane, I know you mean the same. Yet for clarity you might want to write something like Abd/add (frontal plane) or use the same wording in both places.

- *As suggested, we added (abd/add) after the use of 'frontal plane' in text (line 276).*

Line 338-340: Not clear how the peak power was affected by the backlash.

- *As suggested by the reviewer, we clarified how peak power was affected by backlash on lines 395-400: “The R2 condition was predicted by the model to be the best balance between harvesting the most electricity from pendulum oscillations, without damping the pendulum's oscillations to a suboptimal point. However, since the effective damping achieved was less than what was modelled, a condition with higher modelled damping (R5) ended up producing the most electrical power.”*

Line 370-373: How did the step width change with the different harvesting conditions and how is this related (if at all) to the frontal torque?

- *There was no significant difference in step width between conditions as reported in the Results Section 3.4 (lines 291-292). If there had been a difference in the step width, this may have affected the frontal plane interaction torque. This is because wider steps tend to lead to greater trunk M-L input motion, which leads to increases mass oscillation amplitude and frontal plane moments developed by the backpack.*

Appendix D

Dr. Qingguo Li
Bio-Mechatronics and Robotics Laboratory
Department of Mechanical and Materials Engineering, Queen's University
ql3@queensu.ca
ph: 613-533-3191
fax: 613-533-6489

Queen's University
McLaughlin Hall
99 University Ave, Kingston, ON
K7L 3N6
Canada

Dear Dr. Monica Daley,

Please find the enclosed revised manuscript titled "Generating electricity while walking with a medial-lateral oscillating load carriage device". As suggested, we have hosted our study's data on Dryad and have put the data's DOI in the Data Accessibility statement at the end of the manuscript.

Regards,

Dr. Qingguo Li